# Choledochal Cyst Excision in Infants—A Retrospective Study

**DOI:** 10.3390/children10020373

**Published:** 2023-02-14

**Authors:** Adam Kowalski, Grzegorz Kowalewski, Piotr Kaliciński, Katarzyna Pankowska-Woźniak, Marek Szymczak, Hor Ismail, Marek Stefanowicz

**Affiliations:** Department of Pediatric Surgery and Organ Transplantation, Children’s Memorial Health Institute, 04-730 Warsaw, Poland

**Keywords:** choledochal cyst, laparoscopic choledochal cyst resection, choledochal cyst in infants, laparoscopic hepatobiliary surgery

## Abstract

A choledochal cyst is a rare malformation primarily diagnosed in children. The only effective therapy remains surgical cyst resection followed by Roux-en-Y hepaticojejunostomy. Treating asymptomatic neonates remains a point of discussion. Between 1984 and 2021, we performed choledochal cyst (CC) excision in 256 children at our center. Out of this group, we retrospectively reviewed the medical records of 59 patients who were operated on under one year of age. Follow-up ranged from 0.3 to 18 years (median 3.9 years). The preoperative course was asymptomatic in 22 (38%), while 37 patients (62%) had symptoms before surgery. The late postoperative course was uneventful in 45 patients (76%). In symptomatic patients, 16% had late complications, while in asymptomatic patients, only 4%. Late complications were observed in the laparotomy group in seven patients (17%). We did not observe late complications in the laparoscopy group. Early surgical intervention is not followed by a high risk of complications and may prevent the onset of preoperative complications, giving excellent early and long-term results, especially after minimally invasive laparoscopic surgery.

## 1. Introduction

A choledochal cyst is a rare malformation primarily diagnosed in children. With the improvement of access to ultrasound examination, there is a growing number of patients with a choledochal cyst diagnosed during the prenatal or early neonatal period before the onset of symptoms [1,2,3]. The only effective therapy remains surgical cyst resection followed by Roux-en-Y hepaticojejunostomy [4,5]. Treating asymptomatic neonates remains a point of discussion. Usually, the operation is postponed until the 3rd–6th month of life, mainly due to the difficulty of CC resection in neonates and smaller infants. However, several reports suggest that neonatal CC may lead to early liver fibrosis even in asymptomatic patients [6,7]. The introduction of minimally invasive surgery did not conclude the ongoing discussion, as some surgical centers promote early surgery [2], while, in contrast, others postpone the intervention time, even in symptomatic infants, due to the later possibility of robotic surgical access [8,9]. Irrespective of the patient’s age, definitive treatment principles remain the same and consist of complete excision of the cyst and reconstruction of the biliary tract. In this retrospective analysis, we present our experience with surgical treatment of patients with CC who were operated on under one year of age. Our paper aims to assess whether CC resection in young infants is safe, feasible, and might provide additional benefits in the treatment process.

## 2. Materials and Methods

Between 1984 and 2021, we performed choledochal cyst (CC) excision on 256 children at our center. Out of this group, we retrospectively reviewed the medical records of 59 patients operated on under one year of age. All of them underwent total cyst resection with end-to-side Roux-en-Y hepaticojejunostomy. 

The open procedure was based on a right subcostal incision, total cyst resection, creation of a 40 cm length Roux-en-Y jejunal loop, and end-to-side hepaticojejunostomy. Since 2015, a laparoscopic approach for choledochal cyst resection has become our institution’s surgical method of choice. We used a 30° laparoscope introduced to the abdominal cavity through the umbilical incision and three ports of 3–5 mm (Figure 1). Transcutaneous traction sutures through the gallbladder and falciform ligament were used for better liver hilum exposure. A Roux-en-Y loop was created extra abdominally through the widened umbilical incision and returned to the abdomen. The end-to-side anastomosis was performed intraabdominally with interrupted monofilament absorbable 5/0 or 6/0 sutures (Figure 2).

All patients were followed postoperatively at 1, 3, 6 months, and every 12 months after surgery. Physical examination, liver function tests, and ultrasound examinations were retrospectively analyzed.

Standard demographic variables included age, gender, body weight, time of diagnosis, and presence of clinical symptoms during the preoperative period. We analyzed intraoperative, early, and late complications. The patients were subsequently divided into groups based on: the presence of symptoms in the preoperative period (asymptomatic vs. symptomatic), time of diagnosis (prenatal vs. postnatal), and surgical access (laparoscopy vs. laparotomy). We compared patients between groups for their demographic data, laboratory findings, and surgical outcomes. Complications were defined as early (<30 days after CC resection) or late (>30 days after CC resection).

### Statistical Analysis

Statistical analysis was performed using Statistica 13 software. Categorical variables were presented as numbers and percentages. Continuous variables were presented as medians and ranges. Student’s t-test and the Mann–Whitney U test were used to assess unpaired associations between continuous variables. We compared categorical variables using the chi-squared test of independence. A *p*-value of less than 0.05 was considered statistically significant. The study was approved by the Institutional Ethical Committee (approval number: 34/KBE/2021).

## 3. Results

A total of 59 infants, 44 females, and 15 males underwent total cyst resection with end-to-side Roux-en-Y hepaticojejunostomy. Age at surgery ranged from 11 to 345 days (median 116 days), and body weight ranged from 800 g to 10.5 kg (median 5.9 kg). In 20 patients (33%), prenatal ultrasound examination detected an infrahepatic cystic lesion prenatally. The remaining 39 patients had been diagnosed after birth. Of the 59 patients, 26 had a fusiform dilatation of CBD (Todani type 1C) and 33 had a cystic dilatation of CBD (Todani type 1B). The percentage of patients diagnosed prenatally has been rising in our population, from 50% in 2001 to 71% in 2019. Absolute values are shown in Figure 3.

The preoperative course was asymptomatic in 22 (38%) patients, while 37 (62%) had symptoms before surgery. Most commonly, patients presented with prolonged jaundice (29 patients, 49%). Acholic stools occurred in nineteen patients (32%), vomiting occurred in five patients (8%), abdominal mass was present in four patients (6%), and cyst perforation occurred in one patient (2%). Symptomatic patients had higher bilirubin, GGTP, and ALT concentrations. Both groups had no statistical differences between early and late complications after the CC resection. We did not experience any intraoperative complications. A comparison of patients in both groups is shown in Table 1.

Clinical data and demographic characteristics of patients diagnosed prenatally and postnatally are shown in Table 2. Patients diagnosed postnatally were older, had higher transaminase concentration, and had a more frequent symptomatic preoperative course.

Follow-up ranged from 0.3 to 18 years (median 3.9 years). The late postoperative course was uneventful in 45 patients (76%). The results in the general group are presented in Table 3. Two patients required re-operation due to clinically significant anastomotic stricture eight and ten years after primary choledochal cyst resection.

Forty-one patients (69%) were operated on through laparotomy, while in CC excision was performed laparoscopically on eighteen patients (31%). In one case (6%), conversion to laparotomy was required, since a biliary pseudocyst caused by CC perforation was found during laparoscopy. Early postoperative complications were present in seven patients in the laparotomy group (17%), bile leak in five patients (12%), bleeding in one patient (2%), and ileus in one patient (2%). One patient (6%) in the laparoscopy group had both bile leak and ileus. All early complications were resolved surgically with good results. Late complications were observed in the laparotomy group in seven patients (17%). We did not observe late complications in the laparoscopy group, but the follow-up was significantly shorter. Demographic data and a comparison of both groups are shown in Table 4.

## 4. Discussion

The population of patients with early recognized, often asymptomatic CC is rising thanks to the popularization of routine maternal ultrasonography and improvements in diagnostic methods [2,10,11]. The planning of the optimal treatment strategy often needs to manage two opposing interests—i.e., delaying the operation to achieve optimal surgical working space conditions, and introducing treatment before complications of CC occur. In the latter case, holding on, even in asymptomatic patients with CC, may have its price. Diao et al. demonstrated that liver damage of various degrees in patients with asymptomatic CC might affect as much as 75% of patients [2]. Their data were in accordance with the reports from other researchers who suggested that the newborn group should be considered a unique entity of patients with a distinct clinical course and pathology [2,3,7]. Suita et al. suggested that only early surgical intervention can effectively counteract progressive liver fibrosis [7]. Serious adverse events in previously asymptomatic infants included coagulopathy, subdural hemorrhage [12] and other serious bleeding events [13], rapid cyst enlargement along with gastric outlet obstruction [14], cyst rupture [15,16], and production of adhesions leading to accidental injuries during later surgical interventions [10]. In our group, symptomatic patients, apart from the different clinical course, presented augmented markers of hepatic injury and traits of cholestatic disease compared to asymptomatic patients. Surgery was also more challenging due to previous cholangitis or pancreatitis episodes. There were no statistical differences in the postoperative outcome; however, there were more late complications in the symptomatic group than the asymptomatic group (16% vs. 4%). Children in the symptomatic group had an eventful late postoperative period, mainly attributed to episodes of cholangitis. This suggests that operating early, before the symptoms arise, would benefit the patients by removing the risk of increased onset of early or late complications. In our practice, a diagnosis of asymptomatic CC is an indication for a planned surgical operation, usually performed within 1–2 months after the initial investigation. Meanwhile, symptomatic patients qualify more rapidly for surgical intervention, usually after the treatment of cholangitis or pancreatitis. A similar strategy has recently been shared by Tainaka et al., who reported on the safety of laparoscopic procedures in infants [17]. In their protocol, asymptomatic patients qualify for a planned operation, while patients with the symptomatic preoperative course are operated on early in a semi-emergency laparoscopic procedure with excellent results. 

There are also reports describing infants diagnosed with a cystic biliary anomaly with features of CC and biliary atresia. In these cases, early intervention allows one to precisely distinguish between the two entities and introduce the appropriate treatment [18]. Since surgical intervention for biliary atresia should be made early, ideally before 30–45 days of age, delaying treatment would worsen the outcome [19]. 

The risk of CC malignant transformation remains low in the pediatric population but is known to increase with age [4]. Moreover, cases of embryonal rhabdomyosarcoma in children as young as three years with concurrent CC have been reported [20]. Early intervention dramatically improves patients’ survival and treatment outcomes in these cases. 

As previously mentioned, introducing MIS did not resolve the dilemma of operation timing, mainly because laparoscopic CC resection is challenging and the small working space in infants only augments the difficulty. Complications in laparoscopic procedures in some centers include a conversion rate of up to 30%, portal vein thrombosis, and recurrent cholangitis leading to liver transplantation [21]. We did not encounter such problems in our cohort. Robotic surgery, proposed as an easier-to-perform alternative to laparoscopic surgery, has some limitations. Most of the children operated on with this technique were above one year of age [22,23,24]. Some centers prefer to wait until a patient reaches an age where robotic intervention will be technically feasible in the centers using this method. Our experience shows that open and laparoscopic CC resection, even in neonates as young as 11 days, is safe, feasible, and gives excellent long-term results. Anastomotic leakage and strictures, which are the main concerns in open surgery, are considered limited in laparoscopy thanks to the magnified vision and increased accuracy of anastomosis. There are other advantages to laparoscopic procedures, such as an earlier introduction of oral feeding, reduction in pain after surgery, and shorter hospitalization, all of which are widely known [17,25]. Late complications, mainly anastomosis stricture, may become clinically significant and need surgical intervention years after the primary procedure [17,26]. Interestingly, the influence of age at surgery on the formation of anastomotic strictures after CC operation was shown by Kim et al. [27]. In their study, the onset of strictures rose with age, which suggested that repeated inflammation episodes may worsen the surgical outcome. In our cohort, we did not observe this in laparoscopically operated patients. However, this observation requires additional follow-up time to be proven. In our cohort, two patients required re-operation due to an anastomotic stricture eight and ten years after the initial open operation, which underlines the necessity of regular follow-up in these patients. Our own clinical experience and various publications point to another important issue in CC surgery, which is the necessity of a complete resection of the cyst. Leaving the intra-pancreatic portion of the cyst can lead in years ahead to stone formation [28] and risk of carcinoma [29]. During laparoscopic procedures, a complete resection of the intra-pancreatic part was performed, exposing the narrow distal segment of the cyst. The stump was later clipped, ligated, or left unligated in selected cases of a stenotic distal stump, which was previously proven to be safe and feasible by Diao et al. [30].

Our study had several limitations. First of all, the number of cases was relatively small. This, however, is difficult to overcome in a pediatric setting in Europe, especially when taking only infants into consideration. The second limitation was the retrospective nature of the study. Our cohort consisted of patients who underwent CC resection over a period of more than 20 years, which implies the inclusion of patients operated on by various techniques. Third, the follow-up of the laparoscopic group was relatively short. Long-term observation is necessary to observe clinically significant anastomotic strictures and hepatolithiasis. 

## 5. Conclusions

Choledochal cyst resection followed by Roux-Y hepaticojejunostomy is safe and feasible even in neonates and infants. A prenatal diagnosis of a choledochal cyst allows individually planned surgical treatment before complications occur. Early surgical intervention is not followed by the high risk of complications and may prevent the onset of preoperative complications, giving excellent early and long-term results, especially after minimally invasive laparoscopic surgery. Long-term follow-up is crucial in these patients as late complications are possible and may require further surgical treatment.

## Figures and Tables

**Figure 1 children-10-00373-f001:**
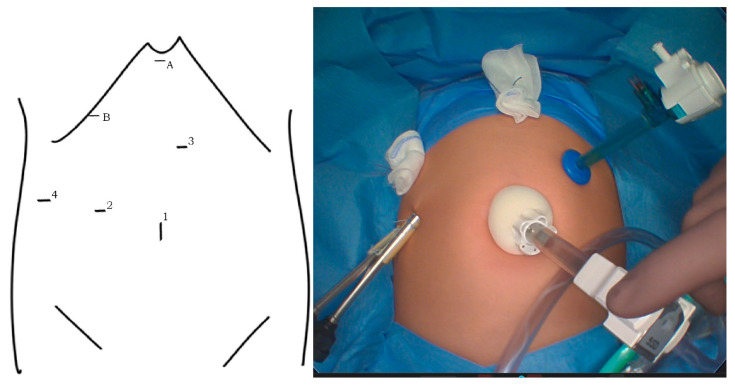
Port placement: 10 mm port for 30° laparoscope in the umbilicus (1), and three device ports of 3–5 mm (2–4). Transcutaneous sutures for the elevation of the falciform ligament (A) and gallbladder (B).

**Figure 2 children-10-00373-f002:**
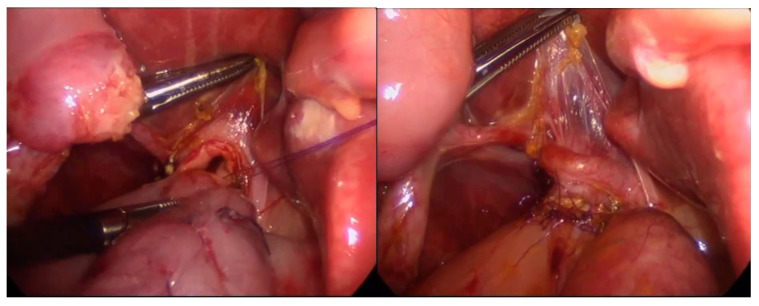
The formation of an end-to-side hepaticojejunostomy with interrupted monofilament absorbable sutures.

**Figure 3 children-10-00373-f003:**
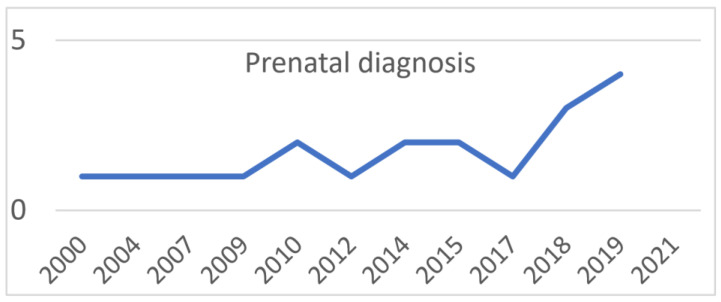
Number of patients with prenatal diagnosis.

**Table 1 children-10-00373-t001:** Clinical data of asymptomatic and symptomatic patients prior to surgery.

	Asymptomatic n = 22	Symptomatic n = 37	*p*
Age (median; range)	75 days; 14–331 days	132 days; 11–346 days	=0.48
Sex (male; no.; %)	5 (23%)	10 (27%)	=0.71
Body weight (median; range)	4.9 kg; 3.1–10 kg	6 kg; 0.8–10.5 kg	=0.77
Total bilirubin mg/dL (median; range)	0.6; 0.1–9.6	7.4; 0.1–19.8	=0.002
ALT U/l (median; range)	21; 7–116	82; 18–452	<0.0001
GGTP U/l (median; range)	120; 16–470	315; 13–2166	=0.037
Time from diagnosis to operation	75 days; 9–325 days	26 days; 2–318 days	=0.14
Early complications (no.; %)	3 (14%)	5 (13%)	=0.99
Late complications (no.; %)	1 (4%)	6 (16%)	=0.18

**Table 2 children-10-00373-t002:** Clinical data of patients diagnosed prenatally and postnatally prior to surgery.

	Prenatal n = 20	Postnatal n = 39	*p*
Age (median; range)	60 days; 11–331 days	152 days; 19–346 days	=0.03
Sex (male; no.; %)	2 (10%)	13 (33%)	=0.07
Body weight (median; range)	4.5 kg; 0.8–10.5 kg	6.35 kg; 3.1–10 kg	=0.35
Total bilirubin mg/dL (median; range)	5; 0.2–19.8	4; 0.1–18	=0.87
ALT U/l (median; range)	29; 7–197	68; 17–452	=0.02
GGTP U/l (median; range)	336; 16–983	223; 13–2166	=0.95
Early complications (no.; %)	4 (20%)	4 (10%)	=0.30
Late complications (no.; %)	2 (10%)	5 (13%)	=0.75
Symptomatic preoperative course	8 (40%)	29 (74%)	=0.0098

**Table 3 children-10-00373-t003:** Postoperative complications.

	All Patients n = 59
Early complications	8 (13%)
Bile leak	6 (10%)
Bleeding	2 (3%)
Ileus	2 (3%)
Late complications	7 (12%)
Cholangitis	7 (12%)
Cholelithiasis	2 (3%)

**Table 4 children-10-00373-t004:** Comparison of patients in open and laparoscopic groups.

	Open n = 41	Laparoscopic n = 18	*p*
Age (median; range)	113 days; 11–346	130 days; 11–331	=0.74
Body weight (median; range)	5.5 kg; 0.8–10.5 kg	6.45 kg; 3.4–9.6 kg	=0.9
Sex (male; no.; %)	11 (27%)	4 (22%)	=0.71
Total bilirubin mg/dL (median; range)	3; 0,1–18	6.9; 0,2–20	=0.70
ALT U/l (median; range)	71; 14–452	33; 7–144	=0.1
GGTP U/l (median; range)	312; 13–2166	174; 16–1020	=0.35
Operating time (median; range)	207; 170–290	222; 135–340	=0.94
Hospitalization days (median; range)	10; 8–20	8; 6–17	=0.029
Follow-up in years (median; range)	6.2; 0.3–18	2.6; 1.4–6	=0.008
Preoperative symptoms (no.; %)	28 (68%)	9 (50%)	=0.18
Jaundice	20 (49%)	9 (50%)	=0.93
Acholic stools	14 (34%)	5 (28%)	=0.63
Vomiting	4 (10%)	1 (5%)	=0.59
Abdominal pain	2 (5%)	1 (5%)	=0.91
Epigastric resistance	3 (7%)	1 (5%)	=0.80
Early complications (no.; %)	7 (17%)	1 (6%)	=0.23
Bile leak	5 (12%)	1 (6%)	=0.44
Bleeding	1 (2%)	0	=0.5
Ileus	1 (2%)	1 (6%)	=0.54
Late complications (no.; %)	7 (17%)	0	=0.06

## Data Availability

All the data are available upon request from the main author at a.kowalski@ipczd.pl.

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
