# Peer review of "Choledochal Cyst Excision in Infants—A Retrospective Study"

_children, 2023, doi:10.3390/children10020373_

Round 1

Reviewer 1 Report

The surgical treatment of choledochal cyst at infants is retrospectively reviewed. Symptomatic surgery tends to have more late complications. Laparoscopic surgery reduces the risk of early and late complications compared to open surgery.

This study is interesting, but there are a few questions and suggestions. 

      Is the title '- is timing important'? As the Introduction indicates, in 'Is  surgery safe and feasible’.

     What is the definition of early and late complications?

     Is it better to add '-at surgery' to the title of each Table? What is the ratio of men to women in each table?

     Table3 is not necessary and if we add that content to Table4.

     About Table 4

     No title.

     Was there any difference in the diameter of the bile duct at the anastomosis? 

     What about the amount of blood loss and length of hospital stay?

What are the preoperative symptoms in the background?

     Please change to text line 103 ‘table 3’, 115 line ‘table 4’.

     This study demonstrates the safety of infants surgery, especially laparoscopy, please add this paperLaparoscopic definitive surgery for choledochal cyst is performed safely and effectively in infants. J Minim Access Surg. 2022 Jul-Sep;18(3):372-377” to the discussion.

Author Response

Dear reviewer,

we are thankful for your interest in our work and all the incisive comments. We tried to answer each one of them thoroughly. Appropriate changes were also made in the manuscript.

Is the title '- is timing important'? As the Introduction indicates, in 'Is surgery safe and feasible’.

We have modified the description of the aim of our work to reflect the intended goal

②     What is the definition of early and late complications?

Complications were defined as early (<30 days after CC resection), or late (>30 days after CC resection). This definition was added in the methods section

③     Is it better to add '-at surgery' to the title of each Table? What is the ratio of men to women in each table?

We added an explanation of time points in our tables

④     Table3 is not necessary and if we add that content to Table4.

Since table 4 is a laparoscopic vs. open procedure comparison, and table 3 shows postoperative complications in the general group, we decided to leave this part of the manuscript unchanged

⑤     About Table 4

     No title.

We added the missing title

     Was there any difference in the diameter of the bile duct at the anastomosis? 

Since this is a retrospective study, unfortunately, we do not have all the available data to make a reliable and statistically sound comparison.

     What about the amount of blood loss and length of hospital stay?

We added hospitalization time to our analysis. The evaluation of blood loss was not assessed in this retrospective analysis.

What are the preoperative symptoms in the background?

All the preoperative symptoms were added to the analysis

⑥     Please change to text line 103 ‘table 3’, 115 line ‘table 4’.

We made the suggested changes

⑦     This study demonstrates the safety of infants surgery, especially laparoscopy, please add this paper “Laparoscopic definitive surgery for choledochal cyst is performed safely and effectively in infants. J Minim Access Surg. 2022 Jul-Sep;18(3):372-377” to the discussion.

The suggested manuscript was added to the discussion part of our paper

Reviewer 2 Report

The authors present a large series of neonates and infants in whom choledochal cyst resection followed by Roux-Y hepaticojejunostomy was performed using minimally invasive and open techniques. Early surgical intervention was not followed by the high risk of complications, and prevented the onset of some preoperative complications giving excellent early and long-term results. Minimally invasive laparoscopic surgery seemed to improve the outcomes, which is in line with the data from quite recent meta-analysis: Sun R, Zhao N, Zhao K, Su Z, Zhang Y, Diao M, Li L. Comparison of efficacy and safety of laparoscopic excision and open operation in children with choledochal cysts: A systematic review and update meta-analysis. PLoS One. 2020 Sep 28;15(9):e0239857. doi: 10.1371/journal.pone.0239857. PMID: 32986787; PMCID: PMC7521726. Some images from the laparoscopic anastomosis formation would be nice to include in the manuscript.

Author Response

We are very grateful for your interest in our work and all the positive comments. We added two images from laparoscopic anastomosis formation according to your suggestions.

Reviewer 3 Report

The authors explained that patients with choledochal cysts were performed laparoscopically and openly, but it would be appropriate to explain how many patients were symptomatic in the laparoscopic group, especially in table 4. Thus, it will be understood in more detail whether late complications develop in the laparoscopic group.

Author Response

Thank you very much for the comment. We listed all the preoperative symptoms in Table 4 according to your suggestions.

Reviewer 4 Report

The authors set up expectations of the study in the title itself: "Is timing important?", but go on to very superficially explore this concept. The matter is not so much about timing but rather about the presence or absence of symptoms of the disease. Moreover, the aim of the study is not phrased in such a way that it reflects the questions asked in the title: "Our paper is aimed to 38 assess whether CC resection in young infants is safe and feasible.".

Please refrain from using any contractions, such as "haven't" or "hasn't".

Please denote decimal points using full-stops instead of commas.

Even though the study is interesting, it is not highly powered enough to detect significant differences between the main comparison of Asymptomatic vs. Symptomatic.

Given the explicit, focused nature of the title, the authors should only have a Table for the main comparison of Asymptomatic vs Symptomatic to help them answer their very focused question. The remainder of the comparative analysis can be limited to text only, with the authors mentioning only any significant differences that they are able to find.

The analysis is univariate and rudimentary and unable to answer the nuanced clinical question that the authors want to answer, i.e., what is the optimal timing of surgery? Even the Asymptomatic vs. Symptomatic comparison is likely marred by confounders, such as the mode of surgery (minimally-invasive vs. open). 

Author Response

The authors set up expectations of the study in the title itself: "Is timing important?", but go on to very superficially explore this concept. The matter is not so much about timing but rather about the presence or absence of symptoms of the disease. Moreover, the aim of the study is not phrased in such a way that it reflects the questions asked in the title: "Our paper is aimed to 38 assess whether CC resection in young infants is safe and feasible.".

Please refrain from using any contractions, such as "haven't" or "hasn't".

Please denote decimal points using full-stops instead of commas.

Even though the study is interesting, it is not highly powered enough to detect significant differences between the main comparison of Asymptomatic vs. Symptomatic.

Given the explicit, focused nature of the title, the authors should only have a Table for the main comparison of Asymptomatic vs Symptomatic to help them answer their very focused question. The remainder of the comparative analysis can be limited to text only, with the authors mentioning only any significant differences that they are able to find.

The analysis is univariate and rudimentary and unable to answer the nuanced clinical question that the authors want to answer, i.e., what is the optimal timing of surgery? Even the Asymptomatic vs. Symptomatic comparison is likely marred by confounders, such as the mode of surgery (minimally-invasive vs. open). 

Dear reviewer,

Thank you very much for your constructive remarks. We modified our manuscript to reflect the changes you suggested. We also changed the title of our manuscript to better correspond with its main goals, expanded the discussion part, and added additional references. You are right to point out that the statistics in our manuscript are too weak to unambiguously answer the question of the appropriate timing of CC surgery. However, it is important to underline that this is a retrospective study, with a relatively small, heterogeneous population. Unfortunately, there is no basis for multivariate analysis in these circumstances. We added a limitations paragraph at the end of the discussion.  We do believe, however, that the study is valuable, interesting and discusses an important clinical question. Moreover, there are not too many papers dedicated solely to the surgery of infants with CC, which makes our work an important contribution to already published data.

Round 2

Reviewer 1 Report

No problem with the revised paper. Accept this paper.